# Effects of Baekje Weir Operation on the Stream–Aquifer Interaction in the Geum River Basin, South Korea

**Hyeonju Lee [1], Min-Ho Koo [2], Byong Wook Cho [1], Yong Hwa Oh [1], Yongje Kim [1]**, 
**Soo Young Cho [1], Jung-Yun Lee [1], Yongcheol Kim [1] and Dong-Hun Kim [1,*]**

[1] Groundwater Research Center, Geologic Environment Division, Korea Institute of Geoscience and Mineral Resources, Daejeon 34132, Korea; hjlee@kigam.re.kr (H.L.); cbw@kigam.re.kr (B.W.C.); yhoh@kigam.re.kr (Y.H.O.); yjkim@kigam.re.kr (Y.K.); sycho@kigam.re.kr (S.Y.C.); 8548@kigam.re.kr (J.-Y.L.); yckim@kigam.re.kr (Y.K.)

[2] Department of Geoenvironmental Sciences, Kongju National University, Kongju 32588, Korea; koo@kongju.ac.kr

\* Correspondence: donghun@kigam.re.kr; Tel.: +82-42-868-3113; Fax: +82-42-868-3414

**Abstract:** Hydraulic structures have a significant impact on riverine environment, leading to changes in stream–aquifer interactions. In South Korea, 16 weirs were constructed in four major rivers, in 2012, to secure sufficient water resources, and some weirs operated periodically for natural ecosystem recovery from 2017. The changed groundwater flow system due to weir operation affected the groundwater level and quality, which also affected groundwater use. In this study, we analyzed the changes in the groundwater flow system near the Geum River during the Baekje weir operation using Visual MODFLOW Classic. Groundwater data from 34 observational wells were evaluated to analyze the impact of weir operation on stream–aquifer interactions. Accordingly, the groundwater discharge rates increased from 0.23 to 0.45 cm/day following the decrease in river levels owing to weir opening, while the hydrological condition changed from gaining to losing streams following weir closure. The variation in groundwater flow affected the groundwater quality during weir operation, changing the groundwater temperature and electrical conductivity (EC). Our results suggest that stream–aquifer interactions are significantly affected by weir operation, consequently, these repeated phenomena could influence the groundwater quality and groundwater use.

**Keywords:** Baekje weir; weir operation; stream–aquifer interaction; Visual MODFLOW

## 1. Introduction

Hydraulic structures such as weirs and dams are constructed in many rivers to control the river flow, affecting the groundwater level near the river. In South Korea, 16 weirs were constructed, and river channels were dredged to secure more water resources in four major rivers (Han, Geum, Yeongsan, and Nakdong rivers), in 2012, which changed the river environments. Groundwater levels rose in most areas near the rivers because river levels increased due to the management level of the weirs [1]. Consequently, weir construction helped to meet water demands and increased flood protection [2–4]. However, concerns about the impact of the restricted rivers on the riparian ecosystem remained [5,6]; therefore, nine weirs have operated periodically for natural ecosystem recovery, from 2017.

Weir operation causes river level fluctuations, facilitating the movement of water and solutes between the river and aquifer [7]. As the river level increases, groundwater levels increase in areas near rivers. In contrast, as the river level decreases, groundwater flows back towards the river, decreasing the groundwater level. In addition, the river level can be further decreased after a weir is opened due

to riverbed dredging during the weir's construction [8,9]. In fact, some agricultural areas suffered from groundwater use during the weir opening in 2017. A change in the stream–aquifer interaction can affect the water quality and biogeochemical cycling [10], which will affect humans who use groundwater. Since 16 weirs were constructed in South Korea, a number of studies have been conducted to evaluate the effect of weir construction on river water or ecosystems [11–15], however, only a few studies have focused on groundwater [16,17]. Therefore, an accurate understanding of the impacts of weir operation on stream–aquifer interaction is important to prepare countermeasures for potential issues and manage water resources efficiently.

Research on stream–aquifer interaction is an extensively studied topic in hydrology, and numerous studies on the impact of hydraulic structures on stream–aquifer interactions have been conducted. Some studies have investigated variations in river levels induced by hydraulic structure changes to the gradient of the hydraulic head between the aquifer and river [18,19], causing a bank storage exchange [20]. Variations in river flow, due to hydraulic structures, can also change riverbed sedimentation [21,22], affecting the river-aquifer connectivity [7]. The hydraulic gradient and riverbed permeability are important factors that determine stream–aquifer interaction [23]. A change in hydrological conditions due to variations in the river level can cause rapid transport of contaminants between an aquifer and a river [24,25], affecting the groundwater quality and aquatic ecosystems [26,27].

In this study, we analyzed the changes in groundwater flow and storage induced by weir operations using Visual MODFLOW Classic (Waterloo Hydrogeologic Inc., v. 4.6.0.169, Waterloo, ON, Canada), a three-dimensional finite difference model, to evaluate the impact of the Baekje weir on the stream–aquifer interaction in the Geum River, South Korea. In parallel, changes in groundwater quality were analyzed to investigate the implications of weir operations on hydrochemical properties.

## 2. Materials and Methods

### 2.1. Study Area and Hydrogeological Setting

The study area of 2.7 km$^2$ alluvial aquifer is located between longitudes 126°56′30′′ E and 126°58′31′′ E, and latitudes 36°18′54′′ N and 36°19′56′′ N in Buyeo near the Geum River, South Korea (Figure 1a). The Geum River flows from the northeast to southwest in the northern part of the study area, and the river width varies from 290 to 570 m. The Baekje weir is located downstream of the study area, and the management level of the weir is 4.2 m. The thalweg elevation near the study area declined by 0.65 m during the riverbed dredging in 2010 [28,29]. Consequently, the depth of the river is approximately 5 m. The Jawangcheon and Sojaengicheon tributaries of the Geum River are located in the eastern and western parts of the study area, respectively. The southern part of the area is characterized by a hilly terrain. The bedrock predominantly consists of granite of the Bulguksa Formation and is unconformably overlain by the Quaternary alluvium. The Quaternary alluvium is distributed along the Geum River and its tributaries. The alluvium, which was deposited in a fluvial environment, is characterized by fining upward sequences, composed of clay, sand, and gravel (Figure 1b). The surface elevation varies from 10 to 15 m, and the average thicknesses of the clay layer and the sand and gravel layer are 11 m and 9.5 m, respectively. The permeable sand and gravel layers are thick near the river, and the thickness decreases with distance from the river. The riverbed of the Geum River is located mainly in the sand and gravel layers.

This area is a greenhouse clustered agricultural area cultivating watermelon and green pumpkin. Groundwater is mainly used for growing crops during the warm season, whereas it is used for water curtain cultivation during the cold season. The study area is typical of greenhouse cultivation areas near weirs in South Korea.

Figure 2 shows the variations of groundwater level in observational well (OW1) located near the river (Figures 1a and 3a) along with the river level during weir operation. The river level declined stepwise during two months from July 2019, due to the opening of the weir gate; this also decreased the groundwater level in the riverside land. After the river level increased owing to the closure of the

water gate, the groundwater level increased again. This result indicates that the groundwater actively interacts with the river water because of weir operation.

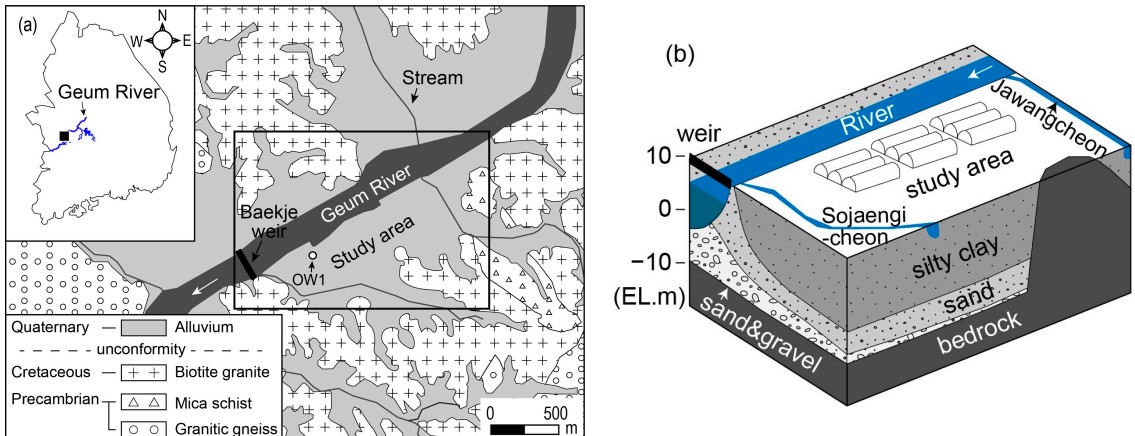

**Figure 1.** (**a**) Location and geological map of the study region in the Geum River Basin, South Korea (modified from [30]); (**b**) Characteristics of the alluvial successions.

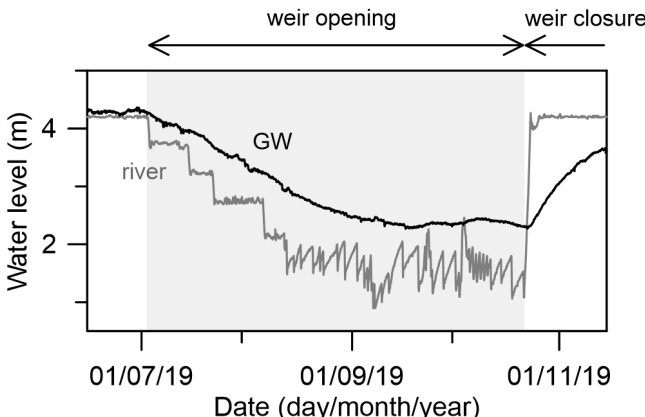

**Figure 2.** Variation of the groundwater level (GW) in observational well (OW1) due to the change in river level.

### 2.2. Groundwater Monitoring Data

Groundwater data were used to evaluate the effect of weir operation on groundwater level and quality. The locations of the 34 groundwater monitoring wells are shown in Figure 3a. Groundwater level was observed in 29 monitoring wells (filled circles in Figure 3a). Field measurements of the groundwater temperature and electrical conductivity (EC) were made, and ion analysis was conducted twice a month on average in the 8 monitoring wells (empty circles in Figure 3a) to study the variations caused by weir operation.

River level data were observed from the National Stream Monitoring Station (NSMS) and used for comparison with the groundwater data. The NSMS data were provided by the Water Management Information System (WAMIS; http://www.wamis.go.kr/ENG). The river water was sampled from the Geum River in July and October 2019 for comparison with groundwater data.

For statistical analysis, the absolute value of the differences between the groundwater and river levels was calculated using Microsoft Excel (v. 2016). Standard box-whisker plots were constructed to analyze the temporal variations of stream–aquifer interactions using Grapher (Golden Software Inc., v. 10, Golden, CO, USA). The box-whisker plots were used for summarizing the distribution of the groundwater monitoring data and comparing the changes over time.

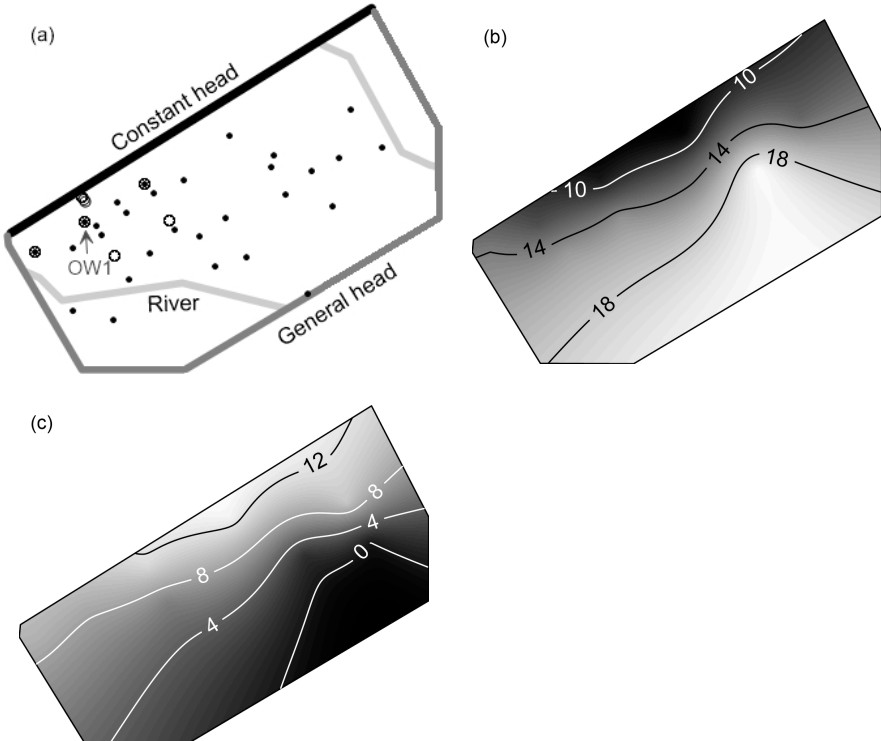

**Figure 3.** (**a**) Conceptual model depicting the boundary conditions and locations of 34 monitoring wells (black circles); (**b**) Depth of the permeable layer; (**c**) Thickness (m) of the permeable layer (gravel layer).

*2.3. Model Development*

Visual MODFLOW was used to simulate the groundwater flow in the alluvial stream-aquifer system undergoing weir operation. Figure 3a shows the model domain with dimensions of 2.7 × 2.3 km, and a grid size of 10 × 10 m was used. The sedimentary structure of the alluvial deposits was reflected in the model using the columnar sections of the 11 monitoring wells. The thick permeable layers are located 10 m below the surface near the river (Figure 3b,c). The depth of the layers increases with distance from the river, while the thickness decreases. The aquifer was vertically discretized into four hydrogeologic layers. The upper layer (mean thickness of 11 m) is composed of silty clay. The second layer consists of sand (mean thickness of 4 m), and the third layer consists of sand and gravel (mean thickness of 5 m). The lower layer is composed of granite.

The Geum River was modeled as a constant head boundary condition. The river level was set to the management level of the Baekje weir (4.2 m). The water levels of Jawangcheon and Sojaengicheon, implemented as river boundary conditions, were set to 7.5 and 6.5 m, respectively. The riverbed conductance was set to 1 m²/day considering the low hydraulic conductivity of the upper layer (silty clay). The hinterland was represented by a general head boundary condition, and its conductance was calibrated. A groundwater recharge (217 mm/year) was applied to the top of the model and calculated from the annual precipitation (1206 mm) and recharge factor (0.18) of the study area [31].

The hydraulic properties of the upper (silty clay) and second (sand) layers were set to 0.01 and 5 m/day, respectively. The hydraulic property of the lower layer (bedrock) was 0.1 m/day [31] and that of the third layer (sand and gravel) was calibrated.

There are 135 pumping wells for water curtain cultivation and 137 for growing crops. The daily pumping rates for each well were set to 30 m³/day for water curtain cultivation and 1 m³/day for growing crops [32]. The pumping rates for water curtain cultivations were estimated from the average yearly groundwater use of each water curtain greenhouse (50 m³/day; [33]), the pumping period (90 days), and the number of water curtain greenhouses per well (2.5 houses).

### 2.4. Model Calibration

The steady-state model was calibrated to groundwater levels from 29 monitoring wells when the river level was 4.2 m (Figure 3a). The calibration was carried out by trial and error, wherein the model parameters were calibrated until the fit between the calculated and observed heads was suitable. The developed model was calibrated by changing the hydraulic conductivity value in the sand and gravel layers, and the conductance of the hinterland.

Several trial and error simulations indicated that the model fit was acceptable between the calculated and observed heads when the hydraulic conductivity was 74 m/day for the sand and gravel layers. The final calibrated conductance of the hinterland was 10 and 20 $m^2$/day for the bedrock and alluvium, respectively. The agreement between the calculated and observed heads is shown in Figure 4. Overall, the calculated heads matched the observed heads reasonably well. The root mean square error (RMSE) between the calculated and observed heads was 0.41 m and normalized root mean square error (NRMSE) was 5.2%. This indicates that the calibrated model is in good agreement with the monitoring data, considering that NRMSE values less than 10% suggest a reliable calibration [34].

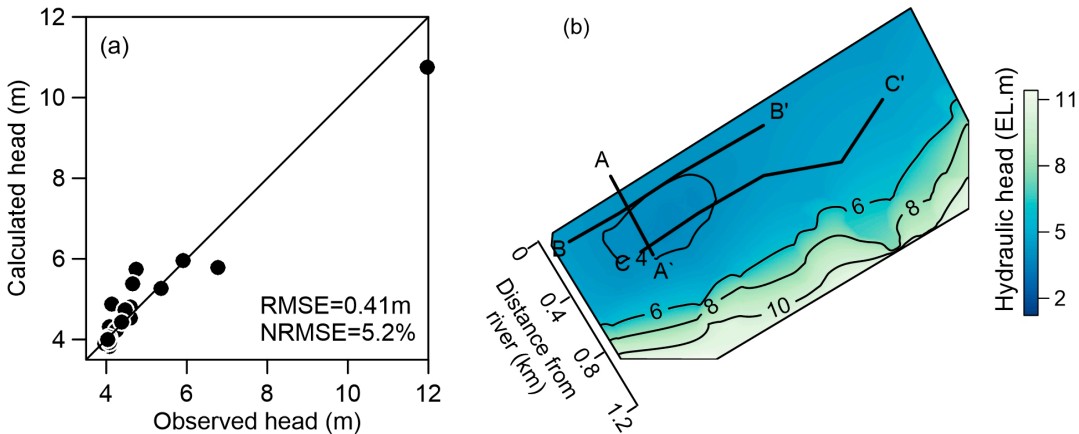

**Figure 4.** Calibration results. (**a**) Comparison of the calculated and observed heads; (**b**) Distribution of the calculated groundwater level.

## 3. Results and Discussion

### 3.1. Change in Stream–Aquifer Interaction

A decline in the river level owing to weir operation changes the hydraulic head gradients, which affects the water flux rates between the aquifer and river [35]. The change in stream–aquifer interactions following the change in river level due to weir operation was investigated using groundwater and river monitoring data. Groundwater levels were observed from 29 monitoring wells for 13 days from 19 June 2019. The average river level was obtained from the NSMS during the same period.

Figure 5 shows the change in the stream–aquifer interaction in the study area. The absolute value of the difference between the groundwater and river levels can be explained as the stream–aquifer interaction. The box-whisker plots, differences between groundwater and river levels, revealed variations of stream–aquifer interaction during weir operations. A positive value implies that the groundwater level is higher than the river level (gaining stream). In contrast, a negative value implies the opposite (losing stream). After a decline in the river level due to weir opening, the groundwater mostly flowed into the river. The median value (the center line of the box) of the difference increased with a decrease in the river level. Following the increase in the river level owing to weir closure, the river level was higher than the groundwater level, implying that gaining streams became losing streams. This indicates that weir-induced river level fluctuations alter stream–aquifer interaction by rapidly changing hydraulic gradients.

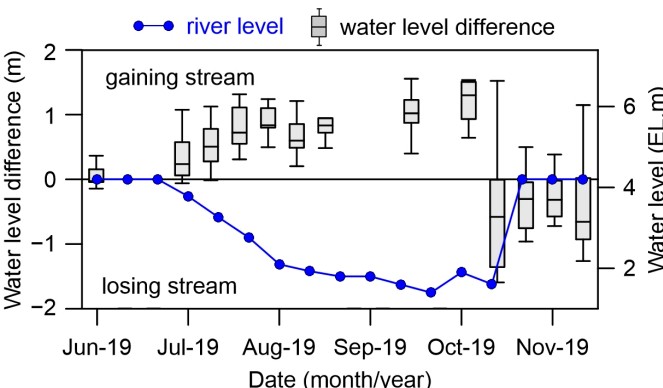

**Figure 5.** Change in groundwater–surface water interaction (box-whisker plots) with river level variation (points) during weir operation.

The model results illustrate that the groundwater level decreases with a decline in the river level, which coincides with the observed values. Figure 6 shows the changes in the groundwater level resulting from weir operation with the sedimentary structure (cross-section line shown in Figure 4b). The groundwater level is higher than the river level (gaining stream) at the management level. After opening the weir, the groundwater level decreased from the upper sand layer to the lower gravel layer in some areas of the riverside land (Figure 6a,b), indicating the decrease in transmissivity. In the protected lowland, the groundwater level was located in the silt layer, even with the decrease in groundwater level, as shown in Figure 6a,c.

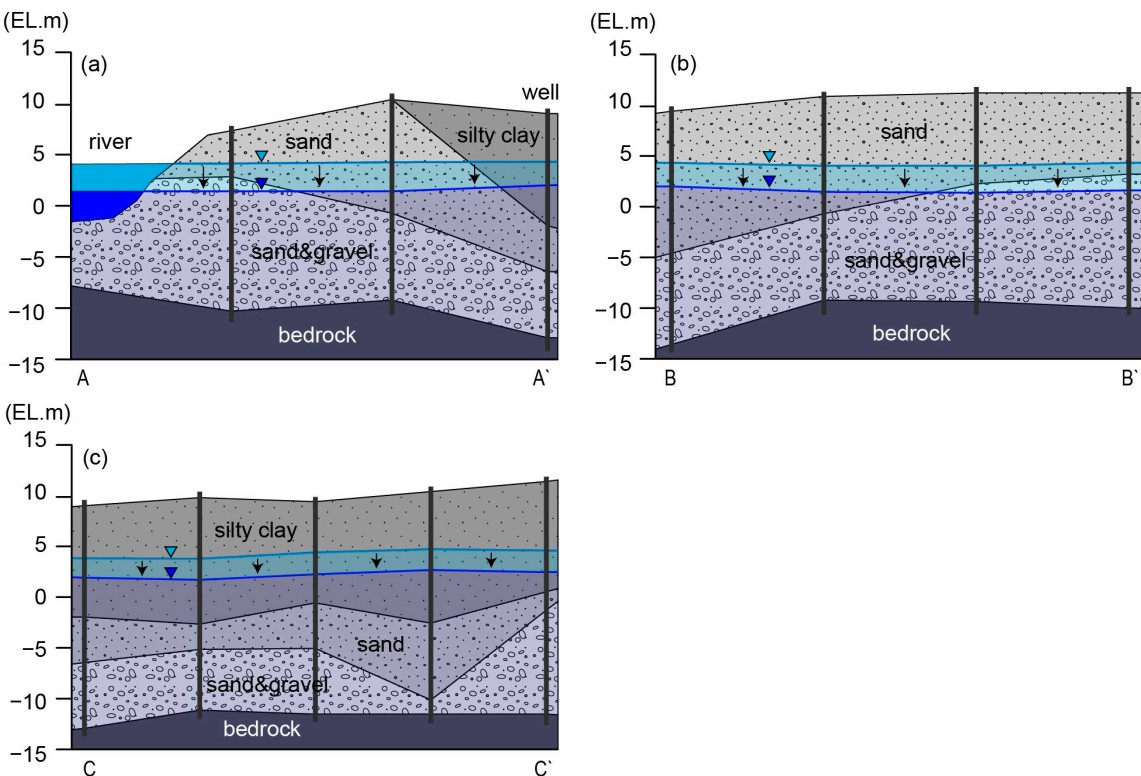

**Figure 6.** Conceptual figure of the groundwater level changes in the alluvial aquifer resulting from weir operation with the geological cross-section. (**a**) A–A'; (**b**) B–B' in the riverside land; (**c**) C–C' in the protected lowland.

The water balance calculated by the simulation shows that the regional groundwater flow from the hinterland to the alluvium of the study area, constituting 47% (2870 m³/day) of the total inflow

(6070 m³/day), increases 1.7 times. The groundwater discharge from the alluvium to the Geum River, constituting 30% (1830 m³/day) of the total inflow, increases 1.9 times (3530 m³/day). The flux rates in the riverbed increase from 0.23 to 0.45 cm/day, which were calculated from the groundwater discharge rates in the riverbed area (0.8 km²). These results indicated that stream–aquifer interactions, due to weir operation, could control the water balance [36]. In addition, the groundwater drawdown could also decrease the discharge rates from alluvium to the tributaries, affecting stream flow rates [37].

　　The impacts of the stream–aquifer interaction due to weir operation on groundwater use were analyzed. Suction-type pumps, usually used for pumping groundwater in the study area, can be used when the groundwater is located within the suction height (8 m). The model assumes that the groundwater is not pumpable if the minimum pumping available thickness (PAT) from the water table to the critical depth is 1 m, considering a further decrease in water level in the wells due to cone of depression and uncertainty in the model [38]. Figure 7 shows the spatial variations of the PAT values during weir operation. Any area with PAT values less than 1 m can be regarded as an area in which groundwater use by suction-type pumps is difficult. When the river level is at the management level of the weir (4.2 m), groundwater can be sufficiently pumped to most of the study area, except for some areas near the river (Figure 7a). However, after the river level decreases (1.4 m), the groundwater level also decreases and the groundwater pumpable region by using suction-type pumps is limited to the hinterland (Figure 7b). In the hinterland, groundwater pumping rates would decrease with a decline in groundwater level, considering the performance characteristics of those types of pumps [39].

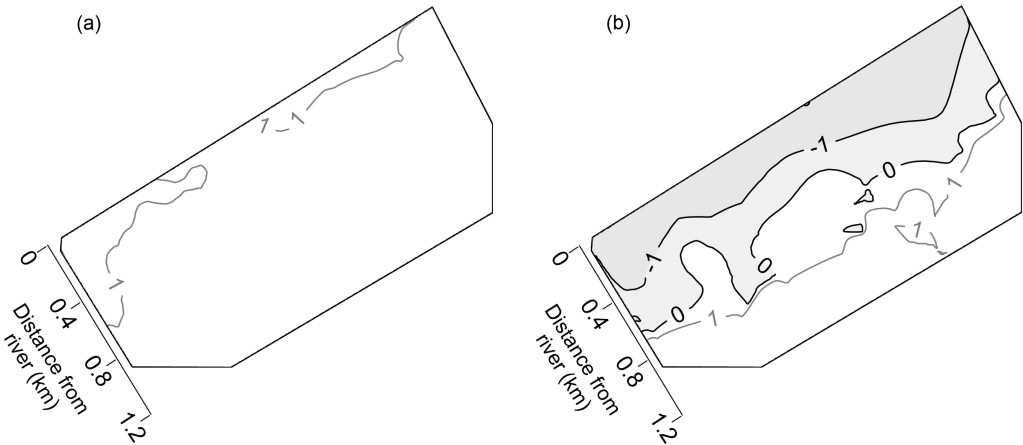

**Figure 7.** Simulated results incorporating the spatial variation of pumping available thickness (PAT) values when the river level is (**a**) 4.2 m and (**b**) 1.4 m, respectively.

### 3.2. Temporal and Spatial Variations in Groundwater Quality

　　As previously discussed, groundwater flow can change owing to weir operation, and consequently, groundwater quality can also be affected [40]. The changed water flux can alter the particle movement, which can deteriorate or improve groundwater quality [41,42]. The impact of weir operations on groundwater quality was evaluated using groundwater physicochemical data from eight monitoring wells which were most significantly affected by river level changes. Figure 8 shows the temporal variation of both the groundwater temperature and EC obtained from the eight monitoring wells. The box height (interquartile range) indicates the degree of dispersion and skewness in the data, and the whiskers indicate the variability outside the upper and lower quartiles. The interquartile range (IQR) of the observed groundwater temperature and EC temporarily increased after the groundwater inflow increased from the hinterland due to a decline in the river level. The large variability of groundwater temperature and EC in the hinterland affected the groundwater near the riverside relatively constantly throughout the year. Both the average values and IQR of the groundwater temperature and EC decreased after weir closure due to mixing of the groundwater and induced cold

river water. The average temperature and EC of river water in October 2019 was 18 °C and 293 µS/cm, respectively, which affected the groundwater.

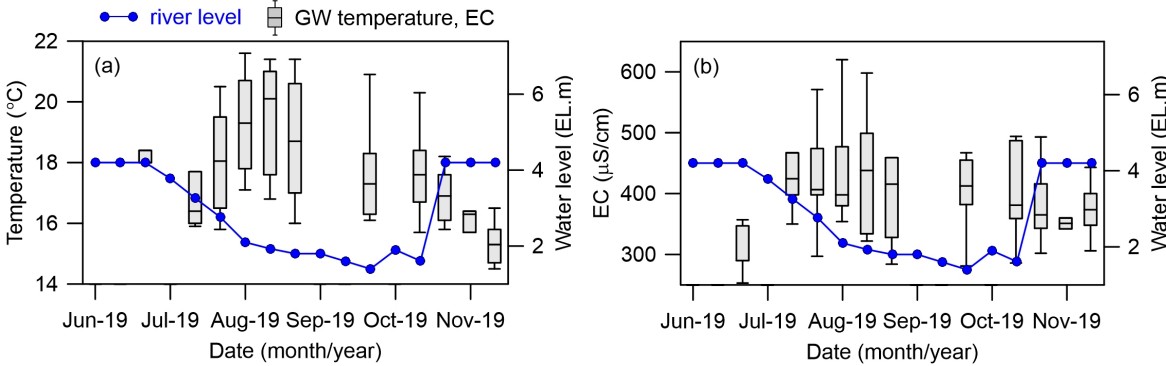

**Figure 8.** Changes in (**a**) groundwater temperature and (**b**) Electrical conductivity (EC) (box-whisker plots) with the river level variation (point) during weir operation.

The modified Piper plot proposed by Chadha [43] was used to analyze the overall ion distribution and groundwater characteristics. Figure 9 shows the changes of the groundwater quality (square) in eight monitoring wells with the river water quality (diamond) in July (after opening the weir) and October 2019 (after the weir closed). From the Chadha plot in Figure 9, the groundwater shows various water types, while the river water is principally the Ca-Mg-Cl type. The groundwater quality is largely controlled by water–rock interactions, local rainfall patterns, and pollution sources [44].

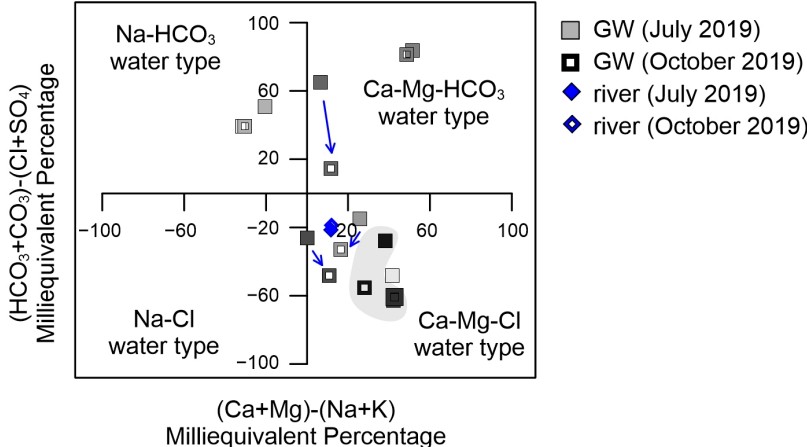

**Figure 9.** Hydrochemical variations due to weir operation in July (after opening the weir) and October 2019 (after weir closure) by Chadha's classification diagram explaining the groundwater (square) and river water (diamond) types.

After the weir opened in July 2019, the groundwater in the riverside land showed the Ca-Mg-HCO₃, Ca-Mg-Cl, and Na-HCO₃ types. The groundwater samples in the protected lowlands were of the Ca-Mg-Cl type, as shown by the gray shadow. After the weir was closed in October 2019, the groundwater quality changed similarly to that of the river water in some areas near the river, including OW1, due to the induced recharge (blue arrows). The groundwater sample closest to the river plotted adjacent to the river water showed a tendency towards river water type even after a month of the weir closure in November 2019, while the groundwater quality in the riverside land, including OW1, returned to its original quality (data not shown). In contrast, protected lowland samples relatively far from the river maintained water qualities despite the weir closing. This indicates that the impact of induced recharge decreases with the distance from the river and the groundwater in the

protected lowland are affected by the surrounding groundwater flow from the hinterland. In addition, the streambed topography also causes localized flow system, showing different changes in groundwater quality near the river [45]. As a consequence of repeated weir operation, the groundwater quality in the riverside land is expected to gradually change and become similar to that of the river, which will affect the groundwater in the protected lowlands. Therefore, continuous long-term monitoring should be conducted in order to understand how the fluctuation of the river level due to weir operation affect the groundwater.

## 4. Conclusions

The Baekje weir was constructed in the Geum River, in 2012, to secure sufficient water resources, and it has operated periodically for natural ecosystem recovery from 2017. The impacts of the Baekje weir operation on the nearby groundwater flow system were analyzed using both numerical simulations and groundwater monitoring data. The conclusions are summarized as follows:

- Variations in the river level owing to weir operation immediately affects the stream–aquifer interaction. A decline in the river level increases the groundwater flux from the aquifer to river by 1.9 times (0.45 cm/day), while an increase in the river level changes the hydrological condition from gaining to losing streams. Consequently, groundwater level changes affect groundwater usability in the agricultural field.
- The change in the stream–aquifer interaction also affects the groundwater quality near the river. The variability of the groundwater temperature and EC are immediately affected by weir opening or closure, causing changes in groundwater quality. This suggests that groundwater quality can be changed by repeated weir operation, consequently, groundwater in the protected area is also affected by the river water.
- Changes in the river environment, such as river channel patterns and river depth due to construction of the weir, can cause unexpected impacts on groundwater during weir operations. Therefore, future study should focus on long-term monitoring of groundwater, as well as changes in connectivity between the aquifer and river affected by weir operations.

**Author Contributions:** Conceptualization, H.L., M.-H.K., and Y.K. (Yongcheol Kim); investigation, H.L., B.W.C., Y.H.O., Y.K. (Yongje Kim), S.Y.C., J.-Y.L., and D.-H.K.; data analysis, H.L.; methodology, H.L. and M.-H.K.; writing—original draft, H.L.; writing—review and editing, H.L., Y.H.O., and D.-H.K.; project administration, Y.K. (Yongcheol Kim); supervision, D.-H.K. All authors have read and agreed to the published version of the manuscript.

**Funding:** This research was supported by the Basic Research Project (20–3411) of the Korea Institute of Geoscience and Mineral Resources (KIGAM) funded by the Ministry of Science and ICT (MSIT).

**Acknowledgments:** We thank the other members of the Groundwater Research Center in KIGAM for their help with data acquisition and sample analysis.

**Conflicts of Interest:** The authors declare no conflict of interest.

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
