# Peer review of "Effects of Baekje Weir Operation on the Stream–Aquifer Interaction in the Geum River Basin, South Korea"

_water, doi:10.3390/w12112984_

Round 1

Reviewer 1 Report

The present form of the manuscript “ Effects of Baekje Weir Operation on the Stream–Aquifer Interaction in the Geum River Basin, South Korea ”  is not ready for technical review. Some section of the manuscript needs improvement. The abstract is poorly written; it needs to be rewritten and should contain introduction aim hypothesis aim result and conclusion. The introduction section is missing in the abstract; one line of the background of study in abstract attracts the reader most. Connective link is missing between different section. Also, the concluding part of the introduction is missing at the end of the introduction. The author should make the introduction section crisp and to the point related to research, which I don't find in the present form of the manuscript.  Material method section seems ok, except the statical test also software use for statical analysis is missing. Similarly, result and discussion also need improvement and require more justification of statement and with some more link with previous studies. There are very few references in the discussion.

Although the study is interesting and could be useful for a certain group of scientific fertanity, therefore, I would suggest improving the manuscript substantially, giving a chance for the next round, because the subject is interesting. However, even an interesting subject does not justify low quality. Especially the introduction and discussion need more attention.

Reviewer 2 Report

Reviewing the manuscript “Effects of Baekje weir operation on the stream-aquifer interaction in the Geum river basin, South Korea" (water-955930) by Hyeonju et al. submitted to Water in September 2020.

This article describes a research of the river-groundwater interaction due to operation of a weir in Geum river (South Korea). The article shows very clearly how the groundwater is affected strongly and immediately by the operation of the weir – increasing and decreasing the water level of the river. The authors discuss the effect of the weir operation on the groundwater level, temperature, EC and even composition (water type).

I find the manuscript interesting because it deals with important issue -the effect of human influence of rivers flow. Moreover, the manuscript was written well, and the authors were checking many aspects of the effect from weir operation. The paper is short and emphasizing results more that discussions, as expected from preliminary research. Therefore, I think that the article should be accepted with minor revision. My remarks are listed below:

General suggestion

  • In section 3.2 (…groundwater quality) there is no discussion but only description of the resuls. I think the authors should discuss about the meaning of the change in water type and that the effect is limited in space (didn't change the water quality of the groundwater in the more distant wells) and what we can conclude from that.
  • Why all the study area is the area before the weir and after it? Is the area after the weir not change? I think it would be interesting to study this area.

substance

  • Lines 102-104 – why groundwater was measured only in 29 wells and not in all the 32 wells?
  • Section 3.2 – I think it a good idea to add in the end of the section that continues monitoring should conducted in order to understand how the fluctuation of the river level due to the weir operation affect the groundwater in the long run – Is the water type of the groundwater returns to its original or if there is a trend?

Figures

  • I think it better to display OW1 in figure 1 in figure 1 (see lines 86 and 103).
  • In Fig. 1b it seems that the eastern tributary is not shown but shown in Fig 3. (I think it would be nice to put their name in the figure).
  • Fig. 3a – near the line of the river (in the north-west area) there are like 3 circles almost at the same point – Are there three different wells close to each other or other reason?
  • Fig. 3b – the image shows the thickness of the gravel layer, but I think it will help the reader to know the depth of the layer (especially since it is not the begin at the surface).
  • Fig. 8 – You should show (and write also in the text) what the temperature ( it is not enough to write "cold river water" – line 230) and EC of the river.

Typing

  • Line 124 – the "2" in "1m2/d" should be up script.

Round 2

Reviewer 1 Report

Good job by the authors. I have below Few things that should be taken consider by the author before the manuscript gets accepted.

  1. Still a scope of improvement in the introduction section. but it's up to author if he wants to work on it.
  2. what property is taken in the error bar? Its to mention inside the text or in the caption.  is it SD SE or CI 95%? I highly recommend taking  95% confidence interval in case the author has taken SD or ER.
  3. The conclusion is very long it should be to the point.

Author should definitely work on second and third points before the final acceptance of the manuscript.

Round 3

Reviewer 1 Report

The author has done substantial work on the manuscript. It does not require further review. I recommend to accept it in the present form.